# Long-Term Survival with Multiple Myeloma: An Italian Experience

**DOI:** 10.3390/cancers17030354

**Published:** 2025-01-22

**Authors:** Francesca Fazio, Martina Gherardini, Elena Rossi, Tommaso Za, Francesca Di Landro, Sonia Morè, Valentina Maria Manieri, Carmine Liberatore, Maria Gabriela Chavez, Velia Bongarzoni, Svitlana Gumenyuk, Maria Grazia Garzia, Miriana Ruggeri, Angela Rago, Mario Biglietto, Luca Franceschini, Valeria Tomarchio, Laura De Padua, Alfonso Piciocchi, Andrea Mengarelli, Alessia Fiorini, Francesca Fioritoni, Massimo Offidani, Valerio De Stefano, Maurizio Martelli, Maria Teresa Petrucci

**Affiliations:** 1Hematology Department of Translational and Precision Medicine, Sapienza University Azienda Policlinico Umberto I, 00185 Rome, Italy; fazio@bce.uniroma1.it (F.F.); martina.gherardini@uniroma1.it (M.G.); martelli@bce.uniroma1.it (M.M.); 2Section of Hematology, Catholic University, Fondazione Policlinico A. Gemelli IRCCS, 00136 Rome, Italy; elena.rossi@unicatt.it (E.R.); toza251978@yahoo.it (T.Z.); francescadilandro92@gmail.com (F.D.L.); valerio.destefano@unicatt.it (V.D.S.); 3Clinica di Ematologia Azienda Ospedaliero, Universitaria delle Marche, 60126 Torrette, Italy; sonia.more@live.it (S.M.); valentina.manieri95@gmail.com (V.M.M.); massimo.offidani@ospedaliriuniti.marche.it (M.O.); 4Hematology Unit, Department of Oncology and Hematology, Pescara Hospital, 65124 Pescara, Italy; carmine.liberatore@asl.pe.it (C.L.); francesca.fioritoni@asl.pe.it (F.F.); 5Divisione di Ematologia, Ospedale Belcolle, 01100 Viterbo, Italy; mariagabriela.chavezorellana@asl.vt.it (M.G.C.); alessiafiorini@libero.it (A.F.); 6Department of Hematology, San Giovanni-Addolorata Hospital, 00184 Rome, Italy; veliabongarzoni@gmail.com; 7Hematology Unit, IRCCS Regina Elena National Cancer Institute, 00144 Rome, Italy; vitana91@yahoo.it (S.G.); andrea.mengarelli@ifo.it (A.M.); 8Department of Hematology, Hematology San Camillo Forlanini Hospital, 00152 Rome, Italy; mgarzia@scamilloforlanini.rm.it; 9Haematology Service, Mazzoni Hospital, 63100 Ascoli Piceno, Italy; mirianaruggeri@gmail.com; 10UOSD Ematologia ASL Roma1, 00193 Rome, Italy; angela.rago@asl.roma1.it; 11Department of Hematology, S.M. Goretti Hospital, Polo Universitario Pontino, 04100 Latina, Italy; m.biglietto97@gmail.com; 12Hematology, Lymphoproliferative Disease Unit, Tor Vergata University Hospital, 00133 Rome, Italy; luca.franceschini@ptvonline.it; 13Unit of Hematology, Stem Cell Transplantation, University Campus Bio-Medico, 00128 Rome, Italy; v.tomarchio@policlinicocampus.it; 14UOC di Ematologia, Trapianto di Cellule Staminali e Terapia Genica, 95123 Frosinone, Italy; laura_dp_81@libero.it; 15Italian Group for Adult Hematologic Diseases (GIMEMA) Data Center, 00182 Rome, Italy; a.piciocchi@gimema.it

**Keywords:** multiple myeloma, outcome, therapy, long-term survival

## Abstract

The overall survival of multiple myeloma patients is in continuous improvement thanks to the availability of novel treatment. The aim of our study is to analyze the clinical profile of multiple myeloma patients who have survived 10 years or longer in order to identify possible predictors of long-term survival.

## 1. Introduction

Multiple myeloma (MM) is a hematologic disease characterized by the malignant proliferation of clonal plasma cells in the bone marrow, associated with an elevated morbidity and mortality due to end-organ damage. Traditionally, (MM) is considered an incurable disease and most patients with MM eventually relapse or become refractory to the available drugs [1]. Treatments for multiple myeloma (MM) have expanded in the last decade, and the introduction of several novel agents, such as immunomodulatory drugs (IMiDs), proteasome inhibitors (PIs), and monoclonal antibodies (moAbs) has markedly improved MM patients’ outcomes in terms of progression-free survival (PFS) and overall survival (OS) [2,3,4]. With the availability of new treatments and the use of high-dose chemotherapy, followed by autologous hematopoietic stem cell transplantation (ASCT), the median OS of newly diagnosed MM (NDMM) patients is around 6–8 years [5,6,7,8]. Before the introduction of ASCT and targeted therapy, the percentage of MM patients alive at 5 years and 10 years from diagnosis was 18% and 2%, respectively [9]. To date, although the 5-year survival rate for patients with MM is reported approximately 50%, some (approximately 28%) patients with MM are still alive at 10 years from diagnosis [10,11]. Furthermore, considering that the improvements in treatment have raised the possibility that MM might be functionally curable and that the OS of MM patients is in continuous improvement, long-term survival with an adequate quality of life should be the goal of treatment. Recent published data show that the clinical behavior of MM may be different in Latin American populations compared to Caucasian or African Americans populations, consistent with a less aggressive course of the disease in Latinos [12]. However, few data have been reported in the literature concerning the clinical characteristics and features of MM patients who have survived 10 years or longer. Previous studies have demonstrated that factors that influence the overall survival of MM patients include specific patient characteristics, disease-related features, and a patient’s response to first-line therapy. Considering clinical and biological multiple myeloma characteristics, previous studies have identified high tumor burden (according to R-ISS and R2-ISS), high-risk cytogenetic abnormalities, and extramedullary multiple myeloma as indicators of poor prognosis [13,14,15,16]. Instead, only a minimal number of observational and retrospective studies have investigated the prognostic role of patient-related features, such as age, frailty, ECOG PS, or comorbidity, on MM patients’ outcome [17,18,19,20]. To fill in the current gap in understanding regarding the prognostic impact of disease and patient-related features, we conducted a retrospective, observational, and multicenter study to analyze the clinical profile of Italian MM patients who have survived 10 years or longer in order to identify possible predictors of long-term survival.

## 2. Materials and Methods

### 2.1. Study Design and Participants

This is a cohort analysis that retrospectively collected and analyzed clinical data on a series of 344 MM patients, diagnosed between February 1986 and October 2012, at 14 hematology centers of central Italy. The data collection period cut-off was identified to allow for adequate follow-up until July 2023. Patients with a documented diagnosis of MM, according to the International Myeloma Working Group (IMWG) criteria, requiring treatment and living longer than 10 years since the beginning of first-line therapy were considered eligible for our study. The clinical data collected at the time of MM diagnosis were age, sex, ECOG PS, hematology parameters, CRAB criteria, organ function, patient’s comorbidity, type of treatment received during the first line and subsequent lines, and use of upfront ASCT. Informed consent was obtained from all individual participants and the study was in accordance with the ethical standard of the institutional national research committee and with the 1964 Helsinki declaration.

### 2.2. Aim of the Study

The aim of our study is to analyze the clinical profile of MM patients who have survived 10 years or longer since the beginning of therapy, in order to identify possible factors influencing long-term survival.

### 2.3. Statistical Analysis

Patients’ characteristics were summarized using relative frequencies for categorical variables and quantiles for continuous variables. Overall survival was estimated using the Kaplan–Meier Product Limit estimator. Differences in terms of OS were evaluated by means of the Log-Rank test or Cox regression model in univariate and multivariate analyses, after assessment of the proportionality of hazards. Hazard ratios (HR) and 95% confidence intervals were reported as parameter results of the Cox regression models. All covariates were evaluated in univariate models and all factors with univariate association with *p*-values < 0.15 were considered in the multivariate models. Backward and stepwise methods were applied to identify the multivariate models with a step-by-step iterative construction that involves the selection of independent variables, to be considered in the final model. All tests were 2-sided, accepting *p* < 0.05 as statistically significant, and confidence intervals were calculated at the 95% level. All analyses were performed using the R software (R Core Team (2023 version). R: A language and environment for statistical computing. R Foundation for Statistical Computing, Vienna, Austria).

## 3. Results

A total of 344 patients with active MM were included in the current study. The clinical characteristics at diagnosis are listed in Table 1. The median age was 59 years (27–83), 194 (56%) were <60 years, and 150 patients (44%) were >60 years; 174 patients (51%) were female. A history of previous monoclonal gammopathy of undetermined significance (MGUS) was present in 109 patients (32%), with a median time of previous MGUS of 50 months (1–396). According to the International Staging System (ISS), 183 patients (53%) were ISS I; 43 patients were not evaluable for ISS. Out of 146 patients evaluable for cytogenetic abnormalities, 22 (15%) had high-risk cytogenetic abnormality, defined as having at least one of the cytogenetic abnormalities related to poor prognosis, which include IGH translocations t(4;14), t(14;16), and t(14;20), del (17p), p53 mutation, and 1q gain/amplification, and 124 (85%) patients had standard-risk cytogenetics. Most patients (72%) were considered eligible for ASCT, according to Italian expert panel recommendations. Considering CRAB criteria, bone lytic lesions were the most common myeloma-defining events (73%), followed by anemia (22%), hypercalcemia (7%), and renal insufficiency (15%). The median bone marrow plasma cells were 35% (IQR 11–95), and 47/272 patients (17%) had an extramedullary disease. Most patients showed IgG-K isotype (56%), followed by IgA-K (17%). The median number of anti-myeloma therapies was 2 (1–10) and most patients (114) had received only 1 line of therapy; 97 patients had received 2 lines of therapy and 133 patients (39%) had received ≥3 lines of therapy. At the time of data cut-off (July 2023), 78 patients (23%) had died, mostly due to disease progression (69%), 91 patients (29%) were still alive and in treatment, and 175 (51%) were in follow-up without treatment. According to the Italian authorization timelines for medicine pricing and reimbursement procedures, 107 patients (31%) had received chemotherapy as first-line therapy and, specifically, most of the patients had received chemotherapy according to VAD scheme (19%) (vincristine, doxorubicin, and dexamethasone) or PAD scheme (6.5%) (bortezomib, doxorubicin, and dexamethasone). Furthermore, most patients had received new drugs as first-line therapy (69%), and the most common combination used was the VTD scheme (bortezomib, thalidomide, and dexamethasone). After first-line therapy, 11% of patients were in sCR, 38% of patients were in CR, 28% were in VGPR, and 18% in PR, with an overall response rate (ORR) of 95%. Additionally, 3% of patients were considered refractory to first-line therapy and 2% of patients were in stable disease. In our cohort, 239 (69%) patients received hematopoietic stem cell transplantation as treatment; specifically, 110/239 (46%) patients had a single autologous transplant, 125/239 (52%) had two autologous transplants, and 4 patients had one autologous and one allogenic transplant. Further, 180 (79%) who received ASCT were given maintenance therapy after first-line therapy, and the most common drug used as maintenance therapy was interferon (48 patients), followed by thalidomide (43 patients). To estimate the impact of maintenance therapy on long-term survival, we calculated the time from the start of the first-line treatment to the initiation of the second-line treatment, exclusively for patients who received maintenance therapy. The median duration observed was 4.7 years (range: 0.1–18.4 years).

Globally, 211 patients (67%) had received one or two lines of therapy, and 133 patients (39%) had instead received ≥3 lines of therapy. Regarding outcomes, after a median follow-up of 13.46 years (11.3–16.3), the median OS was 21.1 years (19.2-NA) (Figure 1). The OS was 52% at 20 years and 46% at 25 years, respectively. According to the lines of therapy, the OS at 20 years was 73% in patients that received 1 or 2 prior lines of therapy, compared to 30% in patients that received 3 or more than 3 lines of therapy (Figure 2). In our cohort, univariate analysis (Table 2) demonstrated that OS was significantly reduced in patients aged older than 60 years, those not eligible for ASCT, those that received more than 1 line of therapy, those with hypoalbuminemia at the time of diagnosis of multiple myeloma, and those that do not receive maintenance therapy after ASCT. We also confirmed the role of ASCT in the prognosis of MM, regardless of the type of induction therapy or the maintenance of post-transplant therapy; specifically, transplant-eligible patients have a better outcome compared to patients not eligible for ASCT (*p* < 0.001). Regarding the number of anti-myeloma therapies received, we observed a statistical difference in terms of OS for patients that received 1 line of therapy and those that received 2 lines of therapy (*p* < 0.001), as well as for patients that received 1 or 2 lines of therapy and those that received ≥3 (*p* < 0.001). In addition, hypoalbuminemia at diagnosis was associated with a significant reduction in terms of OS compared to patients with a normal value of albumin at diagnosis (*p* < 0.001). Instead, in our analysis, obtaining a deeper response after the first-line therapy (sCR-CR vs. VGPR-PR) seems to have no significant impact on OS (*p* = 0.80), and neither does gender (*p* = 0.88), type of therapy (chemotherapy vs. new drugs, *p* = 0.8), previous history of MGUS (*p* = 0.7), or ISS III (*p* = 0.63). The multivariate analysis identified age over 60 years (HR = 1.8 95% CI: 1.02–3.18, *p* = 0.042), hypoalbuminemia at diagnosis of MM (HR = 3.89 95% CI: 0.35–0.82, *p* = 0.004), and a number of anti-myeloma therapies equal to or more than 3 (HR = 3.13 95% CI: 2.12–7.12, *p* < 0.001) as significant independent prognostic factors for reduced OS (Table 3). The prognostic role of ASCT eligibility and of the maintenance therapy after ASCT in terms of prolonged OS was not confirmed in the multivariate analysis. In our cohort, 93 out of 166 (56%) patients that obtained an sCR or CR after a first-line therapy experienced a relapse. Furthermore, more than 50% (54%) of these patients achieved a new CR (11% sCR) after a second-line therapy. In our cohort, a significant difference in terms of time to progression to first relapse was observed in patients that obtained sCR/CR after first-line therapy, compared to patients that obtained VGPR/PR after first-line therapy (6.8 vs. 5.1 years, *p* = 0.004). In addition, patients were grouped based on OS: group A (70), comprising patients who died within 20 years from MM diagnosis; group B (251), comprising patients who have lived for 20 years from MM diagnosis; and group C (23), comprising patients who have lived for more than 20 years. Our comparative analysis shows that the patients in group C have an inferior median age at diagnosis (46 years) compared to patients in group B (59 years) and patients in group A (62) (*p* < 0.001). Patients in group A had a lower level of serum albumin at diagnosis compared to patients in group B and C (*p* ≤ 0.001). The median numbers of previous lines of therapy in the patients in the three groups were 2 (range 1–10; group A), 4 (range 1–9; group B), and 2 (range 1–6; group C) (*p* < 0.001). Specifically, most of the patients in group A received 3 or more lines of therapy (76%), whereas most of the patients in groups B and C received 1 or 2 lines of therapy (71% and 65%, respectively) (*p* < 0.001). In the group A, 60 pts (86%) experienced relapse; in group B, 157 pts (63%) experienced relapse; and in the group C, 13 pts (57%) experienced relapse. Furthermore, the median follow-ups of the three groups were 12.3 years for the patients in group A (9.1, 19.6), 12.2 years for the patients in group B (9.4, 19.8), and 22.3 years for the patients in group C (20.6, 30.2) (*p* < 0.001). The median follow-up of the entire cohort of patients was 13.46 years (11.31; 16.10). Further, 21 out of 23 patients (91%) in group C were considered eligible for ASCT; meanwhile, 183/251 patients (74%) in group B were considered ASCT eligible and 42/70 patients (63%) in group A were considered ASCT eligible (*p* = 0.023). No difference between the three groups was observed in terms of gender, previous history of MGUS, ISS, cytogenetic abnormalities, and the depth of response to first-line therapy. Out of 70 pts in group A, 16 died of causes not related to multiple myeloma, specifically, 4 pts due to COVID19 pneumonia; 3 pts due to myocardial infarction; 3 pts due to stroke; 6 pts due to second neoplasm, and 1 pt due to road accident. Fifty-four of these pts died due to disease progression. Out of the 23 pts who died in group C, 1 pt died due to COVID-19 pneumonia and 22 pts died due to disease progression.

## 4. Discussion

The prognosis of patients with MM has improved over the last decades thanks to the introduction of several novel drugs, and the improvement in survival has been reported in several experiences. Nevertheless, high-dose chemotherapy followed by ASCT is considered the standard of therapy for younger fit patients with newly diagnosed MM, even in the era of novel induction therapy [5]. Before ASCT, the 5- and 10-year OS in MM patients was 18 and 2%, respectively; these figures have increased to 50% and 28%, respectively, in the ASCT era [9,10,11]. In our analysis, conducted on a wide cohort of patients followed for a long period, we have confirmed the favorable prognostic role of ASCT in better outcomes. Indeed, in our experience, ASCT was identified as the single therapeutic variable associated with a significant long-term survival; accordingly, patients receiving ASCT after initial induction treatment have a better outcome compared to patients not eligible for ASCT. According to other prior experience, the type of available front-line treatment before ASCT did not make any difference in terms of long-term survival [12,21]. In fact, in our study, no difference was identified in terms of the type of available front-line therapy when considering long-term survivor patients. In our study, other variables which we found to be associated with long-term OS were age at diagnosis (less than 60 years), serum albumin level at diagnosis (patients with normal values surviving longer), and number of received lines of therapy (patients that received one line of therapy surviving longer). It is known that aging is associated with an increased risk of developing cancer disease and most diagnoses occur in people aged older than 65 years. Additionally, MM biology may differ by age at onset: younger patients showed more favorable prognostic characteristics and fewer adverse prognostic factors, and patients aged less than 50 years [22,23,24] had a significantly longer median OS compared to patients aged older than 50 years. The human aging process is associated with a progressive [25,26,27], irreversible decrease in physiologic reserve, and all these age-related changes can occur in organ function. Furthermore, metabolic changes (such as the pharmacokinetics and pharmacodynamics of drugs) can affect clinical efficacy and can impact on the poor tolerability of anti-myeloma therapy in elderly patients. Reduced drug tolerability negatively impacts the outcome of elderly patients with MM. In our analysis, a significant difference in terms of time to progression to first relapse was observed in patients that obtained sCR/CR after first-line therapy, compared to patients that obtained VGPR/PR after first-line therapy. Nevertheless, obtaining a deeper response after first-line therapy (sCR-CR vs. VGPR-PR) seems to have no significant impact on OS. Taken together, these observations suggest that obtaining a CR is neither a necessary nor sufficient condition for attaining a long survival. Several prospective studies have demonstrated that obtaining a deep response after therapy translates into better outcomes in terms of PFS and OS [28,29]. Nevertheless, it has been demonstrated that, despite not achieving CR, patients with a previous history of smoldering myeloma or patients with a gene expression profiling (GEP) signature according to patients with MGUS can have OS of over 10 years (75% of patients) after HDM-ASCT [30]. Furthermore, a large meta-analysis has established the role of minimal residual disease (MRD) negativity in improving long-term survival in multiple myeloma patients. In recent years, the significance of MRD as a prognostic biomarker has become increasingly clear and, to date, MRD negativity has been adopted as a clinically valid biomarker for outcomes, in terms of PFS and OS, in multiple myeloma [31,32,33,34]. Recently, the FDA’s Oncologic Drugs Advisory Committee accepted MRD as primary endpoint for accelerated approvals in multiple myeloma trials.

In addition, we observed that patients that received more than three lines of therapy and those with hypoalbuminemia at diagnosis of multiple myeloma had a worse outcome compared to patients with a normal value of albumin and those that received one or two lines of therapy. These predictive factors were found in to independent of age at diagnosis be in the multivariable analysis. These data suggest that low tumor burden and slow progressive disease associated with chemosensitivity seem to be more relevant to obtaining a long survival compared to reaching a deeper response, hypothesizing that genetic background and kinetic features could remain stable in time. Unfortunately, the cytogenetic data from our cohort are limited due to the recent availability and real-life application of this test. No molecular or biological characteristics of the disease are available for our cohort of analyzed patients. The heterogeneous pathobiology of multiple myeloma, even within the same patients during the natural history of the disease, has been reported in several studies. These data highlight the genetic and biological chaos underlying multiple myeloma physiopathology, which could be present from the onset of the disease and which evolves rapidly with each subsequent relapse [35,36,37].

To date, despite improvement in the overall survival, mainly related to the introduction of several novel drugs, a limited number of data are available characterizing multiple myeloma patients who have experienced a long survival. Specifically, according to other retrospective studies, in our study, clinical and therapeutic variables are identified as favorable predictors of prolonged OS. Currently, clinical factors orient the choice of therapy in MM patients, but the widespread diffusion and preliminary clinical application of molecular technologies has allowed for the identification of several prognostic and predictive biomarkers for survival outcome. These findings underline the importance of designing newly prospective studies to identify clinical, biological, and molecular characteristics that could be used to better stratify newly diagnosed multiple myeloma patients in order to incorporate reproducible biomarkers [38,39,40] and to identify tailored optimal target therapies.

## 5. Conclusions

Despite the limit of being a retrospective study, the real-life observation of our cohort of long-term survivors with MM identified age of more than 60 years, hypoalbuminemia at diagnosis, and a number of anti-myeloma therapies equal to or more than 3 as significant independent prognostic factors for reduced OS. These finding underline the importance of designing prospective studies to identify clinical, biological, and molecular characteristics that could be used to better stratify newly diagnosed multiple myeloma pts in order to incorporate reproducible biomarkers and to identify tailored optimal target therapies.

## Figures and Tables

**Figure 1 cancers-17-00354-f001:**
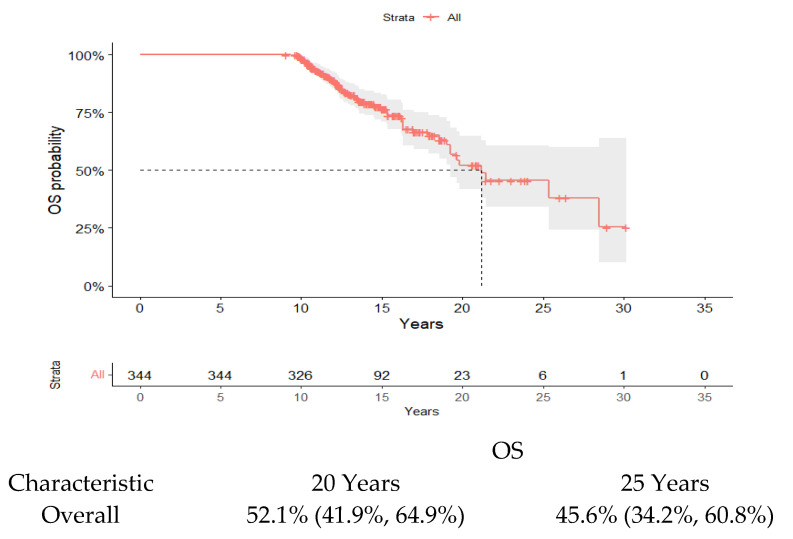
Overall survival of the entire cohort. The median OS was 21.1 years (19.2-NA).

**Figure 2 cancers-17-00354-f002:**
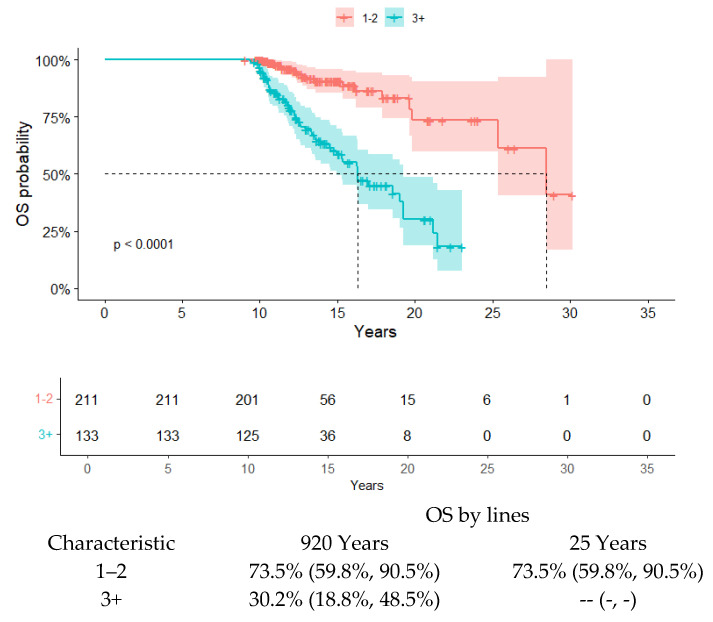
Overall survival of the entire cohort stratified by lines of therapy.

**Table 1 cancers-17-00354-t001:** Patients’ baseline characteristics.

	N = 344
Median age, years (range)	59 (27–83)
Age > 60 years	150 (44)
Male, n (%) Female, n (%)	170 (49) 174 (51)
Previous MGUS, n (%)	109 (32)
ISS I, n (%)	183 (53)
ISS II	67 (19)
ISS III	51 (15)
Unknown	43 (13)
HR cytogenetic abnormalities	22 (15)
Bone lesions, n (%)	250 (73)
Hb (g/dL), median (range)	11.8 (6.6–17.2)
Creatinine (mg/dL), median (range)	0.9 (0.3–3.2)
LDH (U/L), median (range)	264 (82–1375)
Calcemia (mg/dL), median (range)	9.4 (2.2–15.5)
Albumin (g/dL), median (range)	3.92 (2–5.44)
Isotype, n (%)	
IgG-k	192 (56)
IgA-k	57 (16)
IgG-λ	30 (9)
K light chain	20 (6)
λ light chain	17 (5)
IgA-λ	13 (3.5)
Non secretory MM	6 (2)
IgD-λ	4 (1)
IgM-k	2 (0.6)
IgD-k	2 (0.6)
IgM-λ	1 (0.3)
ASCT eligibility, n (%)	246 (72)
Number of anti-myeloma therapy, median (range)	2 (1–10)
Ongoing therapy, n (%)	91 (29)
≥3 lines of therapy, n (%)	133 (39)
Chemotherapy as first line therapy, n (%)	107 (31)
* ORR after first line therapy, (%)	328 (95)
sCR or CR after first line therapy, n (%)	167 (49)
Maintenance therapy after ASCT, n (%)	188/239 (79)

* ORR (overall response rate): patients achieving at least a partial response (≥PR).

**Table 2 cancers-17-00354-t002:** Univariate analysis of OS in relation to clinical characteristics at diagnosis.

	Univariate Analysis		
	HR	95% CI	*p*-Value
Age > 60 years	2.69	1.66–4.35	<0.001
Gender Female	0.97	0.61–1.53	0.88
Previous MGUS	1.08	0.99–1.01	0.7
ISS III	1.2	0.57–2.54	0.63
ASCT eligibility	0.3	0.22– 0.60	<0.001
Lines of anti-myeloma therapy ≥ 2	3.45	1.76–6.75	<0.001
Lines of anti-myeloma therapy ≥ 3	5.16	3.00–8.9	<0.001
New drugs as first-line therapy	1.07	0.64–1.78	0.80
No manteinance vs. manteinance	1.98	1.10–3.24	0.008
VGPR/PR vs. CR/sCR	1.10	0.68–1.79	0.80
SR vs. HR	0.50	0.15–1.68	0.26
Hb at diagnosis	0.94	0.82–1.06	0.30
LDH at diagnosis	1.06	0.87–1.28	0.59
Bone lesions at diagnosis	0.73	0.44–1.22	0.24
Renal insufficiency at diagnosis	0.88	0.76–1.03	0.11
Hypercalcemia at diagnosis	1.04	0.89–1.21	0.65
Hypoalbuminemia at diagnosis	0.45	0.30–0.69	<0.001

**Table 3 cancers-17-00354-t003:** Multivariate analysis of OS in relation to clinical characteristics at diagnosis.

	Multivariate Analysis		
	HR	95% CI	*p*-Value
Age > 60 years	1.8	1.02–3.18	0.016
Lines of anti-myeloma therapy ≥ 3	3.89	2.12–7.12	0.002
Hypoalbuminemia at diagnosis	0.53	0.35–0.82	<0.001

## Data Availability

The original data presented in the study are openly available in a specific tool (RedCap 14.3.13).

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
