# Peer review of "Long-Term Survival with Multiple Myeloma: An Italian Experience"

_cancers, 2025, doi:10.3390/cancers17030354_

Round 1

Reviewer 1 Report

Comments and Suggestions for Authors

This retrospective study aims to evaluate the clinical profiles of multiple myeloma (MM) patients who have survived 10 years or longer to identify the potential predictors of long-term survival. The results highlight some clinical parameters associated with survival. However, there are still some issues to  be addressed:

1. Regarding the maintenance therapy after ASCT, the authors should provide details on the median duration of treatment for a better interpretation of its impact on long-term survival.

2. In Table 2, the HR values for LDH and renal insufficiency at diagnosis are listed as 1, with 95% confidence intervals (CIs) also reported as "1-1." The authors should verify and clarify these results.

3. This study failed to find any parameter for better long-term survival in the multivariate model. How about the effects of maintenance therapy?

4. Since this study focuses on long-term survivors, it would enhance the study to include data on treatment-related adverse events in this cohort to understand the quality of long-term survival. Specifically, are there any ongoing side effects that might affect the quality of long-term survival?

Author Response

Response to reviewers:

Reviewer 1.

  1. Regarding the maintenance therapy after ASCT, the authors should provide details on the median duration of treatment for a better interpretation of its impact on long-term survival.

We thank the reviewer for this insightful comment. To estimate the impact of maintenance therapy on long-term survival, we calculated the time from the start of the first-line treatment to the initiation of the second-line treatment, exclusively for patients who received maintenance therapy. The median duration observed was 4.7 years (range: 0.1–18.4 years). This approach provides insight into the duration of disease control achieved with maintenance therapy and its potential influence on long-term outcomes. We added this information in the manuscript, as you can see highlighted in yellow. (line from 163 to 166).

  1. In Table 2, the HR values for LDH and renal insufficiency at diagnosis are listed as 1, with 95% confidence intervals (CIs) also reported as "1-1." The authors should verify and clarify these results.

We thank the reviewer for pointing this out. To make the results more interpretable, we have recalculated and presented the hazard ratios (HRs) for LDH and renal insufficiency at diagnosis based on an increase of 100 units, rather than 1 unit. This adjustment makes the HRs easier to interpret, while the significance levels remain unchanged. We modified the table 2 in the manuscript.

HR

95%CI

p

Ldh by 100

1.06

0.87, 1.28

0.59

Renal insufficiency by 100

0.88

0.76, 1.03

0.11

  1. This study failed to find any parameter for better long-term survival in the multivariate model. How about the effects of maintenance therapy?

We thank the reviewer for this important observation. We also tested the effect of maintenance therapy, which showed a significant association with overall survival (OS) in the univariate analysis (No maintenance vs maintenance HR: 1.98, 95%CI: 1.19-3.24, p=0.008). However, its significance was lost in the multivariate model, where age, albumin levels, and the number of prior therapy lines emerged as the independent prognostic factors for OS. We added this information in the manuscript text and in the table 2, as you can see highlighted in yellow (line 177 and line 193).

  1. Since this study focuses on long-term survivors, it would enhance the study to include data on treatment-related adverse events in this cohort to understand the quality of long-term survival. Specifically, are there any ongoing side effects that might affect the quality of long-term survival?

We thank the reviewer for the insightful comment. Based on the long follow-up of our cohort, we didn’t collect data, as we do currently, on quality of life. However, according to our clinical data, in our series no treatment-related adverse events other than those expected for the drugs used were described.

Reviewer 2 Report

Comments and Suggestions for Authors

This paper is significant as it presents a multicenter retrospective analysis conducted in Italy, evaluating prognostic factors for survival exceeding 10 years in newly diagnosed myeloma patients. The analysis of 344 cases surviving for over 10 years is particularly noteworthy. However, the inclusion of relatively older cases, while unavoidable due to the nature of the study, may represent a limitation.

1. I recommend that the abstract explicitly include the number of cases analyzed, the range and median years of myeloma diagnosis, as well as the range and median age.

2. The finding that patients with fewer lines of treatment show better survival rates is understandable, as the number of treatment lines reflects, rather than determines, long-term survival. However, it would be important to clarify whether survival rates were influenced by whether patients were diagnosed before or after the introduction of novel therapeutic agents.

3. The observation that initial treatment with novel therapeutic agents did not significantly impact prognosis warrants further discussion. Were the long-term survivors who were diagnosed before the introduction of novel agents primarily those with favorable factors such as fewer high-risk chromosomal abnormalities, higher rates of autologous transplantation, or younger age?

4. Regarding patient age, instead of a binary division, could the survival rate be analyzed by examining its decline across successive age brackets, such as every 10 years?

5. Lastly, the acronym "MM" on line 50 should be changed to "MM."

Author Response

Response to reviewers:

Reviewer 2.

  1. I recommend that the abstract explicitly include the number of cases analyzed, the range and median years of myeloma diagnosis, as well as the range and median age.

We thank the reviewer for this suggestion. I modified the abstract as you request.

  1. The finding that patients with fewer lines of treatment show better survival rates is understandable, as the number of treatment lines reflects, rather than determines, long-term survival. However, it would be important to clarify whether survival rates were influenced by whether patients were diagnosed before or after the introduction of novel therapeutic agents.

We thank the reviewer for this important observation. These results are primarily driven by the selection of patients with at least 10 years of survival, as the study focuses on this subset. Within this cohort, we observed that overall survival (OS) was better for patients who responded well to first-line therapy and did not require subsequent lines of treatment. This finding highlights the critical role of an effective initial response in achieving long-term survival.

  1. The observation that initial treatment with novel therapeutic agents did not significantly impact prognosis warrants further discussion. Were the long-term survivors who were diagnosed before the introduction of novel agents primarily those with favorable factors such as fewer high-risk chromosomal abnormalities, higher rates of autologous transplantation, or younger age?

We thank the reviewer for this thoughtful question. It is important to note that our analysis focuses on a highly selected subset of patients who survived at least 10 years. Within this cohort, we evaluated the use of novel therapeutic agents in both first-line and second-line treatments but did not find statistically significant differences in long-term outcomes. Additionally, it should be considered that this cohort consists of patients whose treatments commenced at least 10 years ago, reflecting the therapeutic landscape of that time.

  1. Regarding patient age, instead of a binary division, could the survival rate be analyzed by examining its decline across successive age brackets, such as every 10 years?

We thank the reviewer for this insightful suggestion. We have analysed the frequency distribution across 10-year age brackets: "0–10, 10–20, 20–30, 30–40, 40–50, 50–60, 60–70, 70–80, 80–90," with the respective frequencies observed as follows: 0, 0, 2, 6, 58, 118, 114, 45, and 1. Considering the observed frequencies and the distribution of treatments (including transplantation) across specific age groups, we determined that the proposed age classification (18-60ys, 60-70ys, 70ys+) is the most appropriate. Furthermore, this classification was also found to be an independent prognostic factor for overall survival (OS). For completeness, we have included the Kaplan-Meier survival curves stratified by age.

  1. Lastly, the acronym "MM" on line 50 should be changed to "MM."

We thank the reviewer for your suggestion. I modified as you request.

Round 2

Reviewer 1 Report

Comments and Suggestions for Authors

The authors have addressed all the major issues adequately. The quality of this manuscript has improved significantly. Only some minor issues are identified.

1. In Table 2, the sum of the percentage of the ISS stage is more than 100%. The authors should fix this issue. Inaddition, the author should also check the percentage of each isotype.

2. In Table 2, please add the unit for each lab parameter.  

Author Response

The authors have addressed all the major issues adequately. The quality of this manuscript has improved significantly. Only some minor issues are identified.

  1. In Table 2, the sum of the percentage of the ISS stage is more than 100%. The authors should fix this issue. In addition, the author should also check the percentage of each isotype.

We thank the reviewer for this insightful observation. I corrected and modified the Table 1 (Patient’s baseline characteristics) in the manuscript, as you can highlighted in yellow.

  1. In Table 2, please add the unit for each lab parameter. 

We thank the reviewer for this suggestion. I corrected the Table 1 (Patient’s baseline characteristics) in the manuscript, and I added the unit for each lab parameter, as you can see highlighted in yellow.